# Reduced-port laparoscopic distal gastrectomy in obese gastric cancer patients

**Dong Yeon Kang, Ho Goon Kim**\*[ʘ], **Dong Yi Kim**[ID]\*[ʘ]

Division of Gastroenterologic Surgery, Department of General Surgery, Chonnam National University Hospital, Gwang-ju, South Korea

[ʘ] These authors contributed equally to this work.

\* dr4477@naver.com (HGK); drkt502@naver.com (DYK)

**Data Availability Statement:** All relevant data are within the paper and its Supporting Information files.

## Abstract

### Background

Reduced-port laparoscopic gastrectomy is currently widely performed for patients with gastric cancer. However, its safety in obese patients has not yet been verified. This is the first study on reduced-port laparoscopic distal gastrectomy (RpLDG) in obese patients with gastric cancer. This study aimed to evaluate the short-term surgical outcomes and investigate the feasibility and safety of RpLDG in obese patients with gastric carcinoma.

### Material and methods

A total of 271 gastric cancer patients who underwent RpLDG at our institution were divided into two groups: non-obese [body mass index (BMI) <30 kg/m2, n = 251; NOG] and obese (BMI ≥30 kg/m2, n = 20; OG). The mean age of the enrolled patients was 64.8 ± 11.4 years, with 72.0% being men and 28.0% women. Operative details and short-term surgical outcomes, including hospital course and postoperative complications, were compared by retrospectively reviewing the medical records.

### Results

No significant difference in operation time was found between the NOG and OG (205.9 ± 40.0 vs. 211.3 ± 37.3 minutes, P = 0.563). Other operative outcomes in the OG, including estimated blood loss (54.1 ± 86.1 vs. 54.0 ± 39.0 mL, P = 0.995) and retrieved lymph nodes (36.2 ± 16.4 vs. 35.5 ± 18.2, P = 0.875), were not inferior to those in the NOG. There were also no statistical differences in short-term surgical outcomes, including the incidence of surgical complications (13.9% vs. 10.0%, P = 1).

### Conclusion

RpLDG can be performed safely in obese gastric cancer patients by an experienced surgeon. It should be considered a feasible alternative to conventional port distal gastrectomy.

**Funding:** The authors received no specific funding for this work.

**Competing interests:** The authors have declared that no competing interests exist.

## Introduction

Gastric cancer is the fifth most commonly diagnosed cancer (5.7%) and the third leading cause of cancer-related deaths (8.5%) worldwide [1]. The incidence of gastric cancer is particularly rising in East Asia. The surgical approach to gastric cancer has also undergone rapid and continuous development. Since its introduction in 1994, laparoscopic gastrectomy has been widely accepted and has become a major option in gastric cancer surgery [2, 3]. Many studies have shown several advantages to laparoscopic gastrectomy over open surgery, even in advanced gastric cancer, and its safety has also been verified [4–6]. In a randomized controlled trial conducted in Korea, the safety and benefits of laparoscopic gastrectomy with D2 dissection were demonstrated in patients with locally advanced gastric cancer [7, 8]. In recent years, many surgeons have focused on minimizing invasiveness. Unlike the conventional laparoscopic approach that uses five ports, a less invasive method utilizing a reduced number of ports, "reduced-port laparoscopic surgery," has been developed. Several studies have shown that reduced-port laparoscopic gastrectomy allows for similar postoperative, oncologic, and 5-year overall survival outcomes to conventional port laparoscopic gastrectomy, while achieving higher cosmetic satisfaction and improved oral intake [9–11]. Our previous study also verified the feasibility and safety of reduced-port laparoscopic gastrectomy [12]. Unfortunately, the suitability of reduced-port laparoscopic gastrectomy in obese patients with gastric cancer is unclear. This is probably because the large amounts of adipose tissue make it difficult to secure the surgical field and perform lymph node dissection without assistance. However, since the benefits of using a reduced number of ports during sleeve gastrectomy for morbidly obese patients have been reported [13, 14], it is necessary to investigate the applicability of this approach to obese patients with gastric carcinoma.

At our institution, reduced-port laparoscopic distal gastrectomy (RpLDG) is offered to all patients with gastric malignancy, regardless of obesity, if radical subtotal gastrectomy and adequate lymph node dissection can be achieved laparoscopically. RpLDG was defined as a surgery using only three ports without assistance, and when additional ports were used, it was considered conversion surgery. The present study aimed to evaluate the short-term surgical outcomes and investigate the feasibility and safety of RpLDG in obese patients with gastric carcinoma at a single institution.

## Material and methods

### Patients

At our institution, since May 2014, reduced-port laparoscopic gastrectomy has been offered to all patients with gastric malignancy if radical gastrectomy and adequate lymph node dissection can be laparoscopically achieved. The indication for reduced-port laparoscopic gastrectomy was clinical stage cTx-3N0. In addition, if a patient suspected of node metastasis wanted laparoscopic surgery, RpLDG was performed limited to cT2N1. Obesity determined according to body mass index (BMI; see below for threshold) was not considered an exclusion criterion. Between August 2014 and April 2020, 314 patients with gastric malignancies underwent reduced-port laparoscopic gastrectomy with regional lymph node dissection. From the initial phase of the study, all obese patients underwent reduced-port surgery. Excluding reduced-port laparoscopic total gastrectomy, a total of 271 patients underwent RpLDG by a single surgeon with experience of ≥60 gastrectomies/year, including both open and laparoscopic surgeries. Before initial data collection for the present study, the surgeon already had experience of a total of more than 500 cases of gastric cancer surgery, of which approximately 70% were laparoscopic surgery.

We divided the patients into two groups based on a threshold BMI of 30 kg/m2; 251 patients with a BMI <30 kg/m2 were classified into the non-obese group (NOG) and 20 with a BMI ≥30 kg/m2 were classified into the obese group (OG). Short-term postoperative outcomes such as surgical findings, course of hospitalization, morbidity, and mortality were analyzed. Bias that may be caused by the learning curve was excluded through the Wald test.

The stomach neoplasm was diagnosed based on pathologic findings after endoscopic biopsy in all patients. Computed tomography (CT), chest radiography, and laboratory testing were performed as routine evaluations before surgery. If imaging studies suggested a high-grade clinical stage and the need to evaluate distant metastasis, positron emission tomography-CT was selectively performed. The distal gastrectomy and regional lymph node dissection procedures followed the Japanese Gastric Cancer Treatment Guidelines [15]. D1+ lymph node dissection was generally performed, with D2 lymph node dissection selected for cases of ≥cT2 tumors or suspected lymph node metastasis. Duet surgery using only three ports without special instruments or assistance was routinely introduced to all patients. If necessary, a port was added or the procedure was converted to conventional port laparoscopic distal gastrectomy. All patients were managed postoperatively based on the principles of Enhanced Recovery After Surgery (ERAS); however, the protocol could not be followed completely due to our hospital limitations. The key ERAS steps included were no use of a nasal-gastric tube, early mobilization, and resumption of oral nutrition from postoperative day (POD) 2. Discharge was recommended on POD 6 to 7, depending on whether the patients fulfilled the discharge criteria.

## Data collection and definition

This retrospective study was approved by the Institutional Review Board of Chonnam National University Hospital, Gwangju, Korea, with a waiver of informed consent (IRB No. CNUH-2020-343). The patients' medical records were reviewed to retrieve clinicopathological characteristics, operative details, and information on the postoperative course. A range of surgical variables, including operation time, estimated blood loss, and number of retrieved lymph nodes as well as postoperative outcomes, including the duration of hospital stay and the incidence of surgical complications, were compared between the two groups.

The pathological staging was based on the seventh edition of the Union Internationale Contre le Cancer/American Joint Committee on Cancer TNM classification system [16]. Postoperative morbidity and mortality were defined as any complications or deaths occurring within 30 days of the operation or during hospitalization. The severity of the complications was graded using the Clavien-Dindo classification [17].

## Statistical analysis

All data were analyzed using Stata/SE version 16.0 (StataCorp., College Station, TX). The normality of distribution of continuous variables was verified using the Kolmogorov-Smirnov test, and these data were expressed as mean and ± standard deviation and compared using the Student's t-test. Categorical variables were expressed as numbers (%) and compared using the two-tailed Fisher's exact test or Pearson's chi-square test. Statistical significance for all analyses was set at P ≤0.05.

## Results

### Patient characteristics

The baseline characteristics of the 271 patients (195 men and 76 women) enrolled in this study are shown in Table 1. The age range was 35–90 years (median ± SD = 64.8 ± 11.4). Only 23

**Table 1. Patient characteristics.**

|  | NOG | OG | Total | P-value |
|---|---|---|---|---|
| No. (%) | 251 (92.6) | 20 (7.4) | 271 (100.0) | |
| Age (years) | 65.0 ± 11.3 | 63.3 ± 12.1 | 64.8 ± 11.4 | 0.517 |
| Sex | | | | 0.079[a] |
| Male | 184 (73.3) | 11 (55.0) | 195 (72.0) | |
| Female | 67 (26.7) | 9 (45.0) | 76 (28.0) | |
| BMI (kg/m2) | 23.4 ± 2.8 | 32.5 ± 2.0 | 24.1 ± 3.6 | 0 |
| Abdominal operation history | | | | 0.392[b] |
| No | 228 (90.8) | 20 (100.0) | 248 (91.5) | |
| Yes | 23 (9.2) | 0 (0.0) | 23 (8.5) | |
| Comorbidity | | | | 0.135[b] |
| No | 81 (32.3) | 3 (15.0) | 84 (31.0) | |
| Yes | 170 (67.7) | 17 (85.0) | 187 (69.0) | |
| ASA physical status | | | | 0.001[b] |
| 1 | 112 (44.6) | 1 (5.0) | 113 (41.6) | |
| 2 | 130 (51.8) | 16 (80.0) | 146 (53.9) | |
| 3 | 8 (3.2) | 3 (15.0) | 11 (4.1) | |
| 4 | 1 (0.4) | 0 (0.0) | 1 (0.4) | |
| No. of tumors | | | | 1[b] |
| 1 | 240 (95.6) | 20 (100.0) | 260 (95.9) | |
| 2 | 10 (4.0) | 0 (0.0) | 10 (3.7) | |
| 4 | 1 (0.4) | 0 (0.0) | 1 (0.4) | |
| Tumor size (mm) | 22.2 ± 19.8 | 26.6 ± 12.8 | 22.5 ± 19.4 | 0.177 |
| Tumor location | | | | 0.208[b] |
| Lower third | 174 (69.3) | 18 (90.0) | 192 (70.8) | |
| Middle third | 68 (27.1) | 2 (10.0) | 70 (25.8) | |
| Upper third | 9 (3.6) | 0 (0.0) | 9 (3.3) | |
| Tumor location | | | | 0.47[b] |
| Ant. wall | 47 (18.7) | 6 (30.0) | 53 (19.6) | |
| Post. wall | 54 (21.5) | 2 (10.0) | 56 (20.7) | |
| Greater curvature | 44 (17.5) | 4 (20.0) | 48 (17.7) | |
| Lesser curvature | 106 (42.2) | 8 (40.0) | 114 (42.1) | |
| Differentiation | | | | 0.485[a] |
| Differentiated | 156 (62.2) | 14 (70.0) | 170 (62.7) | |
| Undifferentiated | 95 (37.8) | 6 (30.0) | 101 (37.3) | |
| Lymphovascular invasion | | | | 0.036[b] |
| No | 229 (91.2) | 15 (75.0) | 244 (90.0) | |
| Yes | 22 (8.8) | 5 (25.0) | 27 (10.0) | |
| Perineural invasion | | | | 0.334[b] |
| No | 237 (94.4) | 18 (90.0) | 255 (94.1) | |
| Yes | 14 (5.6) | 2 (10.0) | 16 (5.9) | |
| Curative | | | | 1[b] |
| R0 | 251 (100) | 20 (100.0) | 271 (100) | |
| R1 | 0 (0.0) | 0 (0.0) | 0 (0.0) | |
| Depth of invasion | | | | 0.147[b] |
| pT1 | 223 (88.8) | 15 (75.0) | 238 (87.8) | |
| pT2 | 13 (5.2) | 2 (10.0) | 15 (5.5) | |
| pT3 | 5 (2.0) | 1 (5.0) | 6 (2.2) | |

*(Continued)*

**Table 1.** (Continued)

| | NOG | OG | Total | P-value |
|---|---|---|---|---|
| **pT4** | 4 (1.6) | 1 (5.0) | 5 (1.8) | |
| **pTis** | 6 (2.4) | 1 (5.0) | 7 (2.6) | |
| **Nodal metastasis** | | | | 0.106 [b] |
| **pN0** | 221 (88.0) | 16 (80.0) | 237 (87.5) | |
| **pN1** | 20 (8.0) | 1 (5.0) | 21 (7.7) | |
| **pN2** | 5 (2.0) | 1 (5.0) | 6 (2.2) | |
| **pN3** | 5 (2.0) | 2 (10.0) | 7 (2.6) | |
| **pTNM stage** | | | | 0.065 [b] |
| **0** | 6 (2.4) | 1 (5.0) | 7 (2.6) | |
| **I** | 225 (89.6) | 15 (75.0) | 240 (88.6) | |
| **II** | 15 (6.0) | 2 (10.0) | 17 (6.3) | |
| **III** | 5 (2.0) | 2 (10.0) | 7 (2.6) | |

NOG, non-obese group; OG, obese group.

BMI, body mass index; ASA, American Society of Anesthesiologists.

[a] Pearson's correlation coefficient.

[b] Fisher's exact test.

patients in the NOG had a history of abdominal surgery. According to the American Society of Anesthesiologists (ASA) physical status classification [18], patients in the OG had a significantly higher score than those in the NOG (ASA ≥2, 55.4% vs. 95%, P = 0.001). Other than ASA physical status, no significant intergroup differences were found with respect to age, sex, abdominal operation history, or comorbidity. All patients underwent curative R0 resection, and pathological TNM (pTNM) stages were evenly distributed. There were no significant differences in histopathological characteristics between the two groups, including number, location, size, and differentiation of tumors.

## Operative details

Wald test was performed to assess the effect of change in skill level over time. A total of 271 patients were divided equally into 9 groups consecutively for statistical analysis, and operation time, which represents skill level, was used as the dependent variable. The Wald test revealed that there was no significant change in proficiency over time. (S1 Fig).

Table 2 summarizes the operative details. There were no differences between the two groups in terms of the operation time and estimated bleeding volume. The techniques of reconstruction and anastomosis were also not significantly different, but as extracorporeal reconstruction was performed more frequently in the NOG, Billroth-I gastrojejunostomy was only performed in the NOG. The extent of lymph node dissection and the number of retrieved lymph nodes did not differ significantly between the two groups.

The results also showed no statistically significant difference in the number of ports used and the port addition rate. Of the 251 patients in the NOG group, 41 (16.3%) required additional ports during surgery. Of these, one trocar was added to 40 patients (15.9%), mostly for counter-traction during D2 dissection and combined resection, such as cholecystectomy and hepatic cyst unroofing, among others. In the remaining patient (0.4%), two trocars were added to perform adhesiolysis at the start of surgery. In the OG group, five patients (25.0%) required four ports; this was due to difficulty in securing the surgical field due to heavy visceral fat in four patients, and because a trocar was added in the fifth patient since laparoscopic ligation of the right gastric artery was impossible due to the mass effect of the cancer.

**Table 2. Operative details.**

| | NOG (n = 251) | OG (n = 20) | Total | P-value |
|---|---|---|---|---|
| Operation time (min) | 205.9 ± 40.0 | 211.3 ± 37.3 | 206.3 ± 39.7 | 0.563 |
| Estimated blood loss (mL) | 54.1 ± 86.1 | 54.0 ± 39.0 | 54.1 ± 83.5 | 0.995 |
| Reconstruction | | | | 0.484[a] |
| Intracorporeal | 219 (87.3) | 19 (95.0) | 238 (87.8) | |
| Extracorporeal | 32 (12.7) | 1 (5.0) | 33 (12.2) | |
| Anastomosis | | | | 0.461[a] |
| B-I | 22 (8.8) | 0 (0.0) | 22 (8.1) | |
| B-II | 213 (84.9) | 19 (95.0) | 232 (85.6) | |
| Roux-en-Y | 16 (6.4) | 1 (5.0) | 17 (6.3) | |
| Nodal dissection | | | | 0.076[b] |
| D1+ | 124 (49.4) | 14 (70.0) | 138 (50.9) | |
| D2 | 127 (50.6) | 6 (30.0) | 133 (49.1) | |
| No. retrieved lymph nodes | 36.2 ± 16.4 | 35.5 ± 18.2 | 36.1 ± 16.5 | 0.875 |
| Omentectomy | | | | 0.418[a] |
| Partial | 12 (4.8) | 0 (0.0) | 12 (4.4) | |
| Complete | 228 (90.8) | 18 (90.0) | 246 (90.8) | |
| Bursectomy | 11 (4.4) | 2 (10.0) | 13 (4.8) | |
| Combined resection | | | | 1[a] |
| No | 235 (93.6) | 19 (95.0) | 254 (93.7) | |
| Yes | 16 (6.4) | 1 (5.0) | 17 (6.3) | |
| No. of ports | | | | 0.394[a] |
| 3 | 210 (83.7) | 15 (75.0) | 225 (83.0) | |
| 4 | 40 (15.9) | 5 (25.0) | 45 (16.6) | |
| 5 | 1 (0.4) | 0 (0.0) | 1 (0.4) | |
| Port addition rate | 16.3% | 25% | 17.0% | 0.351[a] |

NOG, non-obese group; OG, obese group; B, Billroth; ASA, American Society of Anesthesiologists.

[a] Fisher's exact text.

[b] Pearson's correlation coefficient.

To determine the factors affecting the operative details, multivariate regression analysis was performed using several factors, such as patient characteristics and surgical outcomes, as independent variables (Table 3). The results showed that BMI was not correlated with surgery time, whereas estimated blood loss (odds ratio 0.11, 95% confidence interval 0.06–0.17, P = 0.00) and combined resection (odds ratio 25.09, 95% confidence interval 6.07–44.11, P = 0.01) showed significant associations. Operation time was the only significantly associated factor in the analysis of estimated blood loss (odds ratio 0.52, 95% confidence interval 0.26–0.77, P = 0.00).

## Postoperative outcomes

As shown in Table 4, there were no noticeable differences in the hospital course after surgery. The postoperative hospital stays were longer in the OG, but the difference was not statistically significant (7.2 vs. 9.2 days; P = 0.212).

There was no statistically significant difference in the incidence of postoperative complications between the two groups (13.9% vs. 10.0%; P = 1), and there were no mortalities in either group. Severe complications occurred in four patients in the NOG, of which two underwent reoperation. One patient had duodenal stump bleeding, and the other experienced infected

**Table 3. Multivariate logistic regression analysis of operative details.**

| | Operating time | | | | Estimated blood loss | | | |
|---|---|---|---|---|---|---|---|---|
| | Coefficient | 95% CI | | P-value | Coefficient | 95% CI | | P-value |
| | | lower | upper | | | lower | upper | |
| Age | -0.22 | -0.64 | 0.21 | 0.31 | 0.65 | -0.26 | 1.56 | 0.16 |
| Sex | -13.35 | -23.95 | -2.76 | 0.01 | -18.60 | -41.48 | 4.28 | 0.11 |
| BMI (kg/m$^2$) | 6.93 | -10.92 | 24.79 | 0.45 | -5.23 | -43.56 | 33.09 | 0.79 |
| Abdominal operation history | 0.78 | -15.55 | 17.11 | 0.93 | 6.04 | -28.97 | 41.05 | 0.73 |
| Comorbidity | -10.50 | -23.82 | 2.83 | 0.12 | -9.45 | -38.13 | 19.24 | 0.52 |
| ASA physical status | 8.09 | -2.41 | 18.59 | 0.13 | -0.30 | -22.91 | 22.31 | 0.98 |
| Differentiation | 12.47 | 2.43 | 22.52 | 0.02 | -10.46 | -32.21 | 11.29 | 0.34 |
| Depth of invasion | -1.20 | -7.18 | 4.77 | 0.69 | 5.33 | -7.47 | 18.12 | 0.41 |
| Nodal metastasis | -4.17 | -15.62 | 7.28 | 0.47 | 8.76 | -15.80 | 33.31 | 0.48 |
| pTNM stage | 8.55 | -7.41 | 24.51 | 0.29 | 22.76 | -11.43 | 56.94 | 0.19 |
| Operation time | | | | | 0.52 | 0.26 | 0.77 | 0.00 |
| Estimated blood loss | 0.11 | 0.06 | 0.17 | 0.00 | | | | |
| Lymph node dissection | 0.34 | -9.06 | 9.74 | 0.94 | -6.98 | -27.12 | 13.16 | 0.50 |
| No. of retrieved lymph nodes | -0.09 | -0.37 | 0.20 | 0.56 | 0.07 | -0.55 | 0.68 | 0.83 |
| Combined resection | 25.09 | 6.07 | 44.11 | 0.01 | 31.27 | -9.87 | 72.41 | 0.14 |
| No. of ports | 2.99 | -9.56 | 15.54 | 0.64 | 3.76 | -23.17 | 30.69 | 0.78 |

BMI, body mass index; ASA, American Society of Anesthesiologists.

fluid collection at the anastomotic site. The remaining two required extended hospitalization for pancreatitis and ascites. There were no serious complications in the OG, and only two patients suffered from mild ileus and ascites. Multivariate analysis showed no significant risk factor for surgical complications (Table 5).

## Discussion

The continuous developments in laparoscopic technology and increasing experience has meant that an increasing number of surgeons are adopting the minimally invasive approach. Many studies have introduced various methods of less invasive laparoscopic gastrectomy, such as reduced-port, dual-port, and single-port laparoscopic gastrectomy, which have contributed to the feasibility of the procedure [19–22]. However, concerns about adopting these procedures in obese patients remain due to technical difficulties and possible postoperative complications. Moreover, despite their high prevalence of gastric cancer, Asian patients have a relatively lower BMI than Westerners, meaning that there are few studies on obese patients with gastric cancer [23–28]. In addition, all previous studies have focused on conventional laparoscopic gastrectomy, reporting that operation times were significantly longer in obese patients [23–27]. Significant differences in intraoperative blood loss [24] and the postoperative course, including the hospital stay, have also been reported [25, 26]. This study is the first to examine RpLDG in gastric cancer patients with a BMI ≥30 kg/m2, finding no significant difference in surgical and short-term outcomes.

In our study, only the ASA physical status differed between patients in both groups. The observed distribution of ASA physical status in obese patients was different from that in previous studies, probably because the ASA classification system changed significantly in 2014, with BMI ≥30 kg/m2 patients being classified as physical status 2 [18]. In addition, due to ethnic differences, the proportion of the Korean population considered to be obese according to the WHO classification is smaller than that in Western countries [29].

**Table 4. Postoperative outcomes.**

| | NOG (n = 251) | OG (n = 20) | Total | P-value |
|---|---|---|---|---|
| **First flatus (day)** | 3.1 ± 1.0 | 3.1 ± 0.8 | 3.1 ± 1.0 | 0.832 |
| **Water feeding resumption (day)** | 1.0 ± 0.4 | 1.3 ± 0.9 | 1.0 ± 0.4 | 0.292 |
| **Diet resumption (day)** | 2.0 ± 0.5 | 2.4 ± 1.1 | 2.1 ± 0.6 | 0.227 |
| **Hospital stay (day)** | 7.9 ± 4.1 | 9.2 ± 4.9 | 8.0 ± 4.1 | 0.212 |
| **Fever** | | | | 0.421[a] |
| No | 228 (90.8) | 17 (85.0) | 245 (90.4) | |
| Yes | 23 (9.2) | 3 (15.0) | 26 (9.6) | |
| **Transfusion** | | | | 1 [a] |
| No | 245 (97.6) | 20 (100.0) | 265 (97.8) | |
| Yes | 6 (2.4) | 0 (0.0) | 6 (2.2) | |
| **Surgical complication** | | | | 1 [a] |
| No | 216 (86.1) | 18 (90.0) | 234 (86.3) | |
| Yes | 35 (13.9) | 2 (10.0) | 37 (13.7) | |
| **Severity of surgical complication** | | | | 1 [a] |
| Mild | 24 (68.6) | 2 (100.0) | 26 (70.3) | |
| Ileus | 9 | 1 | 10 | |
| Gastric stasis | 5 | 0 | 5 | |
| Ascites | 3 | 1 | 4 | |
| Intraabdominal infection | 4 | 0 | 4 | |
| Wound | 2 | 0 | 2 | |
| Chyle leak | 1 | 0 | 1 | |
| Moderate | 7 (20) | 0 (0.0) | 7 (18.9) | |
| Ileus | 4 | 0 | 4 | |
| Gastric stasis | 2 | 0 | 2 | |
| Ascites | 1 | 0 | 1 | |
| Severe | 4 (11.4) | 0 (0.0) | 4 (10.8) | |
| Bleeding | 1 | 0 | 1 | |
| Intraabdominal infection | 1 | 0 | 1 | |
| Pancreatitis | 1 | 0 | 1 | |
| Ascites | 1 | 0 | 1 | |
| Death | 0 | 0 | 0 | |

NOG, non-obese group; OG, obese group; BMI, body mass index.

[a] Fisher's exact test.

Despite our promising results, we believe that RpLDG in obese patients with gastric cancer is associated with greater technical challenges. The adipose tissue that fills the peritoneal cavity narrows the space available to maneuver, despite the creation of a pneumoperitoneum with appropriate carbon dioxide pressure. The operator also needs to retract the fatty and heavy omentum with their left hand during RpLDG, making the procedure more labor intense.

Some authors have reported that laparoscopic gastrectomy in obese patients takes longer than that in non-obese patients [14, 26, 27]. However, in our study, the operation time showed no significant difference between the two groups (205.9 ± 40.0 vs. 211.3 ± 37.3 min, P = 0.563), and the multivariate regression analysis showed that BMI was not correlated with operation time. This may be related to the intracorporeal anastomosis. In obese patients, it is difficult to perform the anastomosis through an epigastric mini-laparotomy because of the thick abdominal wall. Kim et al. [30] reported that obese gastric cancer patients who underwent

**Table 5. Multivariate logistic regression analysis of surgical complications.**

| | Odds ratio | 95% CI | | P-value |
|---|---|---|---|---|
| | | lower | upper | |
| Age | 1.00 | 0.97 | 1.04 | 0.82 |
| Sex | 0.74 | 0.30 | 1.78 | 0.50 |
| BMI (kg/m2) | 0.94 | 0.19 | 4.60 | 0.94 |
| Abdominal operation history | 0.23 | 0.03 | 1.88 | 0.17 |
| Comorbidity | 0.86 | 0.30 | 2.52 | 0.79 |
| ASA physical status | 0.79 | 0.32 | 1.95 | 0.61 |
| Differentiation | 0.91 | 0.41 | 2.01 | 0.82 |
| Depth of invasion | 0.16 | 0.01 | 1.96 | 0.15 |
| Nodal metastasis | 1.18 | 0.33 | 4.24 | 0.80 |
| pTNM stage | 1.32 | 0.03 | 50.07 | 0.88 |
| Operation time | 1.00 | 0.99 | 1.01 | 0.88 |
| Estimated blood loss | 1.00 | 0.99 | 1.01 | 0.58 |
| Lymph node dissection | 1.59 | 0.75 | 3.38 | 0.23 |
| No. of retrieved lymph nodes | 1.00 | 0.98 | 1.02 | 0.95 |
| Combined resection | 1.00 | 0.24 | 4.13 | 1.00 |
| No. of ports | 2.04 | 0.85 | 4.92 | 0.11 |

BMI, body mass index; ASA, American Society of Anesthesiologists.

extracorporeal anastomoses had longer operation times and more postoperative complications than those who underwent intracorporeal anastomoses. For this reason, we also performed intracorporeal anastomosis in 19 of 20 obese patients. Only one obese patient underwent extracorporeal anastomosis due to difficulty in ligating the right gastric artery because of the mass effect of the tumor, ultimately necessitating an additional port and small epigastric incision.

Obese patients have a large amount of visceral fat, which is brittle and bleeds easily, requiring cautious and meticulous handling while exposing vessels. Several studies have reported a longer operation time and greater intraoperative blood loss with laparoscopic gastrectomy in obese patients [24–26]. However, in the current study, the estimated bleeding volume did not differ between the two groups (54.1 ± 86.1 vs. 54.0 ± 39.0 ml, P = 0.995). We believe that the combination of selecting an appropriate energy source and using of short-pitch dissection played an important role in achieving these results. In addition, by performing surgery solely through a reduced-port approach, unexpected bleeding may have been prevented by eliminating inappropriate manual manipulation.

Reducing vapor, which obscures the surgical field, is also an important factor in shortening the operation time. The large amount of adipose tissue in the abdominal cavity of obese patients contains a significant volume of water, generating more vapor from the action of the sealing device. We preferred to use the LigaSure™ (Valleylab, Boulder, CO) over the Harmonic Ace® (Ethicon Endo-Surgery, Cincinnati, OH) in obese patients as it seemed to create less vapor residue on the laparoscopic camera. Therefore, the number of pneumoperitoneum desufflations in the OG was not significantly different from that in the NOG, resulting in comparable operation time between the two groups.

Intracorporeal anastomosis not only shortened the operation time but also had a positive effect on the postoperative course. Several studies have reported that extracorporeal anastomosis in obese patients with gastric cancer leads to inferior early surgical outcomes. Kim et al. [25,

30] reported that obese patients who underwent extracorporeal anastomosis had longer times to bowel recovery, longer hospital stays, and more postoperative complications, including wound problems. Chen et al. [26] reported that totally laparoscopic gastrectomy is more feasible than laparoscopic-assisted gastrectomy, especially in obese patients, with faster recovery and fewer complications. At our institution, 19 out of the 20 OG patients underwent intracorporeal anastomosis, and we assumed that this would allow for equivalent postoperative courses in the two groups. Accordingly, the logistic regression analysis identified no risk factors for surgical complications. Although evaluation of several other factors affecting the postoperative course could not be performed, we believe that intracorporeal anastomosis reduced the negative effect of obesity on short-term surgical outcomes.

Several limitations to this study need to be acknowledged. First, there was a large difference in the number of patients between the two groups. The number of patients with a BMI $\geq$30 kg/m2 was very small. According to the WHO Asian-Pacific guidelines, a BMI $\geq$25 kg/m2 or higher could be used as a standard threshold for defining obesity but we did not consider this classification in our analyses to ensure that this procedure was safe even in more serious conditions. Given the small number of patients with obesity, the results of this study should be interpreted with caution. In addition, to eliminate the inevitable bias inherent to retrospective studies, we limited the analysis to procedures performed by a single experienced surgeon. However, this also makes it difficult to generalize the results of our study.

## Conclusions

RpLDG can be a safe alternative to conventional port distal gastrectomy in obese patients and is applicable for the treatment of advanced as well as early gastric carcinoma. Our experience has shown that the indication for RpLDG can be extended to obese patients. However, we still believe that reduced-port laparoscopic surgery is difficult for several reasons; hence, it should be performed by well-trained, and experienced surgeons. Furthermore, the clinical benefits of reduced-port laparoscopic gastrectomy in obese patients must be validated in a large, multi-center, randomized clinical trial.

## Supporting information

**S1 Fig. Wald test results among 271 patients.**
(TIF)

## Author Contributions

**Conceptualization:** Dong Yeon Kang, Ho Goon Kim, Dong Yi Kim.

**Data curation:** Dong Yeon Kang, Ho Goon Kim.

**Formal analysis:** Dong Yeon Kang, Ho Goon Kim.

**Investigation:** Dong Yeon Kang, Ho Goon Kim.

**Methodology:** Dong Yeon Kang, Ho Goon Kim.

**Project administration:** Dong Yi Kim.

**Supervision:** Dong Yi Kim.

**Writing – original draft:** Dong Yeon Kang.

**Writing – review & editing:** Ho Goon Kim, Dong Yi Kim.

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
