## [Decision Letter · Decision Letter 0]

5 Jul 2021

PONE-D-21-19851

Reduced-port laparoscopic distal gastrectomy in obese gastric cancer patients

PLOS ONE

Dear Dr. Kim,

Thank you for submitting your manuscript to PLOS ONE. After careful consideration, we feel that it has merit but does not fully meet PLOS ONE’s publication criteria as it currently stands. Therefore, we invite you to submit a revised version of the manuscript that addresses the points raised during the review process.

Please address the issues and revise accordingly.

We look forward to receiving your revised manuscript.

Kind regards,

Academic Editor

PLOS ONE

Journal Requirements:

Reviewers' comments:

Reviewer's Responses to Questions

**Comments to the Author**

1. Is the manuscript technically sound, and do the data support the conclusions?

Reviewer #1: Yes

Reviewer #2: Partly

2. Has the statistical analysis been performed appropriately and rigorously? 

Reviewer #1: Yes

Reviewer #2: Yes

3. Have the authors made all data underlying the findings in their manuscript fully available?

Reviewer #1: Yes

Reviewer #2: Yes

4. Is the manuscript presented in an intelligible fashion and written in standard English?

Reviewer #1: Yes

Reviewer #2: Yes

5. Review Comments to the Author

Reviewer #1: In this research article, the authors aimed to evaluate the short-term surgical outcomes and investigate the feasibility and safety of reduced-port laparoscopic distal gastrectomy in obese patients with gastric carcinoma. They conducted a retrospective study of 271 gastric cancer patients who underwent reduced-port laparoscopic distal gastrectomy, where they divided the patients into two groups as non - obese and obese in order to compare and analyse their surgical outcomes. They concluded that reduced-port laparoscopic distal gastrectomy could be performed safely in obese gastric cancer patients. The study is well designed and adequately written. I congratulate the authors for their successful work.

Reviewer #2: Comments to the manuscript:

I think that it is an interesting paper to reflect the preliminary results of reduced-port laparoscopic gastrectomy in patients with gastric malignancy (with distal gastrectomy), comparing obese patients with non-obese patients. However, as the authors reflect in the limitations of the study, it is a retrospective study with few patients, so I would accept it in a short and more summarized format.

I further state some considerations that I believe should be better clarified in this manuscript:

1. In the Methodology section, what were the exclusion criteria for the study? The authors state that this surgery was offered to all patients a priori, but they do not state whether patients were initially excluded for reasons other than obesity. It should also be specified at what tumor stage this surgery was offered to patients with gastric neoplasia. It is only mentioned that surgery was considered when it was technically feasible, but it should be better detailed in cases where this surgical approach was not considered.

2. On the other hand, other aspects that seem to be better in this surgical approach with respect to the conventional laparoscopic approach could have been evaluated, such as cosmetic satisfaction and early oral intake. In addition, tumor recurrence and mean follow-up time are variables that I consider important to analyze.

3. Given the previous studies, which reflect a longer surgical time in obese patients with conventional laparoscopic approach, it would be interesting to compare, as a future project, obese patients with conventional laparoscopic approach versus reduced-port laparoscopic gastrectomy.

6. PLOS authors have the option to publish the peer review history of their article (what does this mean?). If published, this will include your full peer review and any attached files.

Reviewer #1: No

Reviewer #2: No

---

## [Author Response · Author response to Decision Letter 0]

12 Jul 2021

We are thankful for your favorable reception of our research. As you suggest, we corrected the manuscript and attached a file named 'Response to Reviewers'. I hope you feel that revised manuscript meet PLOS ONE's publication criteria.

---

## [Decision Letter · Decision Letter 1]

26 Jul 2021

Reduced-port laparoscopic distal gastrectomy in obese gastric cancer patients

PONE-D-21-19851R1

Dear Dr. Kim,

We’re pleased to inform you that your manuscript has been judged scientifically suitable for publication and will be formally accepted for publication once it meets all outstanding technical requirements.

Kind regards,

Academic Editor

PLOS ONE

Additional Editor Comments (optional):

Reviewers' comments:

Reviewer's Responses to Questions

**Comments to the Author**

1. If the authors have adequately addressed your comments raised in a previous round of review and you feel that this manuscript is now acceptable for publication, you may indicate that here to bypass the “Comments to the Author” section, enter your conflict of interest statement in the “Confidential to Editor” section, and submit your "Accept" recommendation.

Reviewer #1: All comments have been addressed

Reviewer #2: All comments have been addressed

2. Is the manuscript technically sound, and do the data support the conclusions?

Reviewer #1: Yes

Reviewer #2: Yes

3. Has the statistical analysis been performed appropriately and rigorously? 

Reviewer #1: Yes

Reviewer #2: Yes

4. Have the authors made all data underlying the findings in their manuscript fully available?

Reviewer #1: Yes

Reviewer #2: Yes

5. Is the manuscript presented in an intelligible fashion and written in standard English?

Reviewer #1: Yes

Reviewer #2: Yes

6. Review Comments to the Author

Reviewer #1: In this research article, the authors aimed to evaluate the short-term surgical outcomes and investigate the feasibility and safety of reduced-port laparoscopic distal gastrectomy in obese patients with gastric carcinoma. They concluded that reduced-port laparoscopic distal gastrectomy could be performed safely in obese gastric cancer patients. The study is well designed and adequately written. Necessary revisions have been carried out.

Reviewer #2: (No Response)

7. PLOS authors have the option to publish the peer review history of their article (what does this mean?). If published, this will include your full peer review and any attached files.

Reviewer #1: No

Reviewer #2: No

---

## [Editor Report · Acceptance letter]

28 Jul 2021

PONE-D-21-19851R1 

Reduced-port laparoscopic distal gastrectomy in obese gastric cancer patients 

Dear Dr. Kim:

I'm pleased to inform you that your manuscript has been deemed suitable for publication in PLOS ONE. Congratulations! Your manuscript is now with our production department. 

Kind regards, 

on behalf of

Dr. Robert Jeenchen Chen 

Academic Editor

PLOS ONE